# Depth of Response to Intensive Chemotherapy Has Significant Prognostic Value among Acute Myeloid Leukemia (AML) Patients Undergoing Allogeneic Hematopoietic Stem-Cell Transplantation with Intermediate or Adverse Risk at Diagnosis Compared to At-Risk Group According to European Leukemia Net 2017 Risk Stratification

**DOI:** 10.3390/cancers14133199

**Published:** 2022-06-29

**Authors:** Tong-Yoon Kim, Silvia Park, Daehun Kwag, Jong-Hyuk Lee, Joonyeop Lee, Gi-June Min, Sung-Soo Park, Young-Woo Jeon, Seung-Hawn Shin, Seung-Ah Yahng, Jae-Ho Yoon, Sung-Eun Lee, Byung-Sik Cho, Ki-Seong Eom, Yoo-Jin Kim, Seok Lee, Chang-Ki Min, Seok-Goo Cho, Jong-Wook Lee, Hee-Je Kim

**Affiliations:** 1Department of Hematology, Catholic Hematology Hospital, Seoul St. Mary’s Hospital, College of Medicine, The Catholic University of Korea, Seoul 06591, Korea; tyk@catholic.ac.kr (T.-Y.K.); silvia.park@catholic.ac.kr (S.P.); kdh@catholic.ac.kr (D.K.); jonglee@catholic.ac.kr (J.-H.L.); lommu1@catholic.ac.kr (J.L.); beichest@catholic.ac.kr (G.-J.M.); sspark@catholic.ac.kr (S.-S.P.); royoon@catholic.ac.kr (J.-H.Y.); lee86@catholic.ac.kr (S.-E.L.); cbscho@catholic.ac.kr (B.-S.C.); dreom@catholic.ac.kr (K.-S.E.); yoojink@catholic.ac.kr (Y.-J.K.); leeseok@catholic.ac.kr (S.L.); ckmin@catholic.ac.kr (C.-K.M.); chosg@catholic.ac.kr (S.-G.C.); jwlee@catholic.ac.kr (J.-W.L.); 2Leukemia Research Institute, College of Medicine, The Catholic University of Korea, Seoul 06591, Korea; 3Department of Hematology, Yeouido St. Mary’s Hospital, College of Medicine, The Catholic University of Korea, Seoul 06591, Korea; native47@catholic.ac.kr; 4Department of Hematology, Eunpyeong St. Mary’s Hospital, College of Medicine, The Catholic University of Korea, Seoul 06591, Korea; chironhmt@catholic.ac.kr; 5Department of Hematology, Incheon St. Mary’s Hospital, College of Medicine, The Catholic University of Korea, Seoul 06591, Korea; saymd@catholic.ac.kr

**Keywords:** acute myeloid leukemia, allogeneic transplantation, ELN 2017 risk classification, prognosis

## Abstract

**Simple Summary:**

Acute myeloid leukemia (AML) is a devastating but potentially curable disease. The updated version of the European Leukemia Net (ELN) 2017 genetic risk stratification is used as the standard for the prognosis and classification of AML. In the present study, we evaluated the prognostic value of the ELN 2017 criteria on post-hematopoietic stem-cell transplantation (HSCT) outcomes and compared it with pre-HSCT measurable residual disease (MRD) status, determined by Wilms tumor gene 1 (*WT1*) expression. We classified the patients as intermediate (INT) risk and adverse (ADV) risk. We found that the ELN 2017 risk classification did not effectively predict post-HSCT outcomes in patients with INT or ADV risk. The pre-HSCT *WT1* level predicted post-HSCT relapse better than ELN 2017 and had a more prominent prognostic value in the ELN INT risk group than in the ADV risk group.

**Abstract:**

We evaluated the prognostic efficiency of the European Leukemia Net (ELN) 2017 criteria on the post-transplant outcomes of 174 patients with intermediate (INT; *n* = 108, 62%) or adverse (ADV) risk (*n* = 66, 38%) of acute myeloid leukemia; these patients had received the first allogeneic hematopoietic stem-cell transplantation (HSCT) at remission. After a median follow-up period of 18 months, the 2 year OS, RFS, and CIR after HSCT were estimated to be 58.6% vs. 64.4% (*p* = 0.299), 50.5% vs. 53.7% (*p* = 0.533), and 26.9% vs. 36.9% (*p* = 0.060) in the INT and ADV risk groups, respectively. Compared to the ELN 2017 stratification, pre-HSCT *WT1* levels (cutoff: 250 copies/10^4^ ABL) more effectively segregated the post-HSCT outcomes of INT risk patients compared to ADV risk patients regarding their 2 year OS (64.2% vs. 51.5%, *p* = 0.099), RFS (59.4% vs. 32.4%, *p* = 0.003), and CIR (18.9% vs. 60.0% *p* < 0.001). Indeed, high *WT1* levels were more prominent in INT risk patients than in ADV risk patients. Notably, *FLT3*-ITD had the greatest impact on post-HSCT outcomes among all the ELN 2017 criteria components; patients in the *FLT3*-ITD mutant subgroups exhibited the worst outcomes regardless of their allelic ratios or *NPM1* status compared to the pre-HSCT *WT1* level of other INT and ADV risk patients.

## 1. Introduction

Acute myeloid leukemia (AML) is a devastating but potentially curable disease with a 5 year relative survival rate of up to 50% for younger patients [1,2]. With the increasing knowledge of the genomic landscapes that underly AML pathogenesis, there have been substantial changes in the classification of AML, and the prognostication system has evolved [3,4,5,6]. At present, the updated version of the European Leukemia Net (ELN) 2017 genetic risk stratification [7] is adopted as the standard for AML prognostication, with the wide application of next-generation sequencing (NGS).

AML patients are allocated into three possible risk groups by the revised ELN, on the basis of certain cytogenetic or molecular aberrations at diagnosis, which serves as a basis for the establishment of treatment strategies. Since ELN 2017 stratification primarily represents the inherited risk of AML at diagnosis, it cannot explain disease kinetics. In the same context, it is unclear whether this genetic risk stratification is applicable for outcome prediction in the setting of allogeneic hematopoietic stem-cell transplantation (HSCT) although it was not intended to stratify post-HSCT outcomes. The issue has already been addressed in earlier research by German researchers, where they found that the ELN 2017 classification remains effective even after HSCT [8]. However, patients receiving nonmyeloablative (NMA)-conditioning protocols comprised 71.1% of the cohort in this study, which is unlikely to represent the totality of people receiving AML transplants. As conditioning regimens and alternative donor selection strategies differ between continents [9,10], it is debatable whether the ELN2017 risk could predict HSCT outcomes in other cohorts.

Regarding the various aspects of risk prediction during treatment, mounting evidence suggests that invisible disease burden estimated by minimal (measurable) residual disease (MRD) pre or post HSCT has a significant impact on the recurrence of the disease [11,12,13,14,15,16,17]. Although active discussions regarding the best tool for MRD assessment have been ongoing, multiparameter flow cytometry (MFC), polymerase chain reaction (PCR), and NGS are all known to reliably detect MRD [18,19,20]. However, to date, owing to difficulties in standardization, the routine use of these methods in clinical practice presents challenges. Wilms tumor gene 1 (*WT1*) is a potential marker for MRD assessment, with broad applicability, comparability of results, and a standardized assay with a validated cutoff level [21]. The *WT1* gene is located on chromosome 11p13 and encodes the zinc finger transcription factor involved in the regulation of the cell cycle, proliferation, differentiation, and apoptosis [22]. It has been shown to be highly expressed in a majority of hematologic malignancies, including AML, and, therefore, has been intensively studied as a potential marker for minimal residual disease [23,24,25,26,27,28,29,30]. Its prognostic impact on AML is consistent across different study groups, and our institution has used *WT1* expression for a long time [31,32,33,34,35,36,37], even though current recommendations do not favorably support the use of *WT1* for MRD assessment in this era of novel technologies [18,20].

Considering this, we evaluated the prognostic value of ELN 2017 criteria for post-HSCT outcomes and compared them with the value of pre-HSCT MRD status, determined by *WT1* expression.

## 2. Materials and Methods

### 2.1. Patient Selection

The cohort consisted of 174 patients with an intermediate (INT) risk or adverse (ADV) risk of AML according to the ELN 2017 recommendations, who underwent allogeneic HSCT at Seoul St. Mary’s Hospital between November 2017 and November 2020. All patients were treated with standard anthracycline plus cytarabine-intensive chemotherapy and underwent consolidation treatment with allogeneic HSCT after achieving complete remission (CR) or CR with incomplete peripheral recovery (CRi). Figure 1 demonstrates the CONSORT flow diagram for patient selection. This study was approved by the Institutional Review Board and Ethics Committee of the Catholic Medical Center in South Korea (KC21RISI0572).

### 2.2. Preparative Regimens for Allogeneic HSCT

Conditioning regimens were chosen according to the institutional strategy in consideration of the donor type, age, or comorbidities of the patient at the time of HSCT. As for myeloablative conditioning (MAC) protocols, BuCy (busulfan 3.2 mg/kg/day for 4 days from D-7 to D-4 and cyclophosphamide 60 mg/kg/day for 2 days from D-3 to D-2) for matched sibling donor (MSD) or matched unrelated donor (MUD) transplantation, and FluBu2TBI800 (total body irradiation (TBI) 400 cGy/day for 2 days on D-9 to 8, fludarabine 30 mg/m^2^/day for 5 days from D-7 to D-3, busulfan 3.2 mg/kg/day for 2 days from D-6 to D-5) for haploidentical donor (HID) transplantation have been the mainstay of regimens for younger patients. CyTBI (cyclophosphamide 60 mg/kg/day for 2 days from D-7 to D-6 and TBI 330 cGy/day for 4 days from D-5 to D-2) or FluAraCTBI (fludarabine 30 mg/m^2^/day for 5 days from D-9 to D-5, cytarabine 3g/m^2^/day for 3 days from D-9 to D-7, and TBI 400 cGy/day for 3 days on D-4 to D-2) regimens were also used, while the latter was only applied for patients receiving cord blood transplantation. For elderly or fragile patients, a reduced-intensity conditioning (RIC) protocol, FluBu2TBI400 (fludarabine 30 mg/m^2^/day for 5 days from D-6 to D-2, busulfan 3.2 mg/kg/day for 2 days from D-5 to D-4, and TBI 400 cGy/day for 1 day on D-1), was primarily chosen, regardless of donor type. The prevention of graft-versus-host disease (GVHD) differs according to the status of HLA matching and the type of transplant donor. Cyclosporine (transplantation from MSD) or tacrolimus (transplantation from other donors) with methotrexate was used for GVHD prevention. Additionally, patients received mycophenolate mofetil at 3 g three times per day if receiving umbilical cord blood transplantation. Anti-thymocyte globulin (ATG) was used for patients with an unrelated donor (1.25 mg/kg once daily for 2 days, D-3 to D-2) or haploidentical donor (1.25 mg/kg once daily for 4 days, D-4 to D-1), while it was not routinely administered to patients with a matched sibling donor.

### 2.3. Detection of Genetic Mutation on Pretreatment Bone Marrow Samples

In patients with adequate available samples, genomic DNA was extracted from the bone marrow (BM) aspirates with the QIAamp DNA Mini Kit (Qiagen, Hamburg, Germany). NGS analysis was performed using a customized St. Mary’s customized NGS panel for acute leukemia (SM acute leukemia panel) including 67 genes. Target capturing sequencing was performed using a customized target kit (3039061, Agilent Technologies, Santa Clara, CA, USA), and library preparation was carried out according to the manufacturer’s instructions. NGS was performed on an Illumina HiSeq4000 platform (Illumina, Inc., San Diego, CA, USA). Sequenced reads were mapped to the human reference genome (hg19, Genome Reference Consortium, February 2009).

For detection of FMS-like tyrosine kinase 3-internal tandem duplication (*FLT3*-ITD) mutation, PCR for fragment analysis was performed using a modified protocol as previously described [38,39,40,41]. The functional domains of the *FLT3* gene (Gene Bank Accession NM_004119.2) were PCR-amplified with forward primers that were 5′ end-labeled with a fluorescent dye. The PCR products were analyzed using a model 3130XL genetic analyzer (Applied Biosystems, Foster City, CA, USA), and the amplicons with a size greater than that of the wildtype (328 ± 1 base) were considered positive for the ITD mutation. The number, area, and length of mutant peaks on the electropherogram were analyzed using GeneMapper analysis software (Applied Biosystems). The *FLT3*-ITD mutant allelic burden was calculated as the ratio of the area under the curve of mutant and wildtype alleles (mutant/total *FLT3*).

### 2.4. Assessment of WT1 Transcript Levels

The expression of BM *WT1* before allo-HSCT was determined via real-time quantitative-PCR using the *WT1* ProfileQuant kit (Ipsogen, Marseille, France) [21]. *WT1* gene transcripts generated by RQ-PCR were normalized with respect to the number of Abelson murine leukemia viral oncogene homolog 1 (ABL1) transcripts and expressed as copy numbers per 10^4^ copies of ABL1. The cutoff level was 250 copies of *WT1* per 10^4^ copies of *ABL1* [21], specifically, ≥250 copies for *WT1*^high^ and <250 copies for *WT1*^low^.

### 2.5. Definition of Outcomes and Statistical Analyses

Baseline clinical, demographic, and molecular features between the patients with INT risk and ADV risk by ELN 2017 were compared using the chi-square test or Fisher exact test for categorical variables, and a two-sample *t*-test or Mann–Whitney *U* test for continuous variables. Overall survival (OS), relapse-free survival (RFS), the cumulative incidence of relapse (CIR), and non-relapse mortality (NRM) were calculated from the date of allogeneic transplantation. OS was defined as death from any cause. For RFS, relapse and death, whichever occurred first, were considered uncensored events. For CIR, relapse was assessed as an uncensored event, and death in CR was considered a competing cause of failure. NRM was defined as death with relapse as a competing risk factor. The OS and RFS were estimated using the Kaplan–Meier method, and groups were compared using the log-rank test. The Cox proportional hazards model was used for univariate and multivariate analysis for OS and RFS. CIR and NRM were estimated in a competing risk framework using the cumulative incidence of competing events, Gray test for univariate analysis, and Fine–Gray proportional hazard regression for multivariate analysis. Variables with a *p*-value < 0.10, determined by univariate analysis, were considered for entry into the multivariate analysis, while ELN 2017 risk, as a variable of interest, was included in the multivariate analysis, regardless of the *p*-value from univariate analysis. Statistical analyses were performed using R statistical software (version 3.4.3; R Foundation for Statistical Computing, Vienna, Austria).

## 3. Results

### 3.1. Patient Characteristics

In the 174 AML patient cohort, on the basis of the ELN 2017 risk stratification, 108 and 66 patients were allocated to the INT risk (62%) and ADV risk (38%) groups, respectively (Table 1). The median age at transplantation was 54 years (range, 18–74 years), and 37.9% of patients had a hematopoietic stem-cell transplantation-specific comorbidity index (HCT-CI) of ≥3 at the time of transplantation. Most patients included in this study received MAC transplants (*n* = 138, 79.3%). HID transplants were the most common and accounted for 43.7%, which were followed by transplantations from MUD (26.4%), MSD (25.9%), and mismatched unrelated donors or umbilical cord blood (4%). The remission status at allogeneic HSCT was CR1 in 90.8% and CR2 in 9.2% of patients. Pre-transplantation *WT1* levels were available in all patients; 48 were *WT1*^high^ (27.6%) and 126 were *WT1*^low^ (72.4%) at allogeneic HSCT.

There were no significant differences in age, sex, HCT-CI, disease type, donor type, and pre-HSCT *WT1* level between the INT and ADV risk patients. However, more patients in the ADV risk group received MAC transplants compared to the INT risk group (87.9% vs. 74.1%, *p* = 0.047).

### 3.2. Post-HSCT Outcomes According to ELN 2017 Genetic Risk Stratification

The clinical outcomes after transplantation are presented in Figure 2. The probability of OS and RFS at 2 years after allogeneic HSCT for the entire cohort were 60.7% (95% CI, 53.6–68.9) and 51.8% (95% CI, 44.6–60.1). Specifically, for patients of the INT and ADV risk groups, the 2 year OS and RFS were estimated to be 58.6% vs. 64.4% (OS, *p* = 0.299, Figure 2A) and 50.5% vs. 53.7% (RFS, *p* = 0.533, Figure 2B), respectively. The CIR and NRM at 2 years after allogeneic HSCT were 31.2% (95% CI, 23.9–38.8) and 22.3% (95% CI, 16.0–29.3) for the entire cohort: in detail, 26.9% vs. 36.9% (CIR, *p* = 0.060, Figure 2C) and 29.0% vs. 19.6% (NRM, *p* = 0.015, Figure 2D) for patients with INT- and ADV-risk by ELN, respectively.

The results of univariate and multivariate analysis are demonstrated in Table 2. ELN 2017 risk did not show a significant prognostic value for post-transplant OS, RFS, and CIR in the univariate and multivariate analyses, whereas pre-HSCT *WT1*^high^ was independently associated with worse OS (*p* = 0.02, hazard ratio (HR) = 1.81, (95% confidence interval (CI) 1.09–3.02)), RFS (*p* = 0.005, HR = 1.92, (95% CI 1.21–3.03)), and higher risk of CIR (*p* < 0.001, HR = 4.22, (95% CI 2.43–7.81)) after transplantation. Regarding NRM, age ≥ 55 years (*p* = 0.002, HR = 3.90, (95% CI 1.66–9.15)) and ELN ADV risk (*p* = 0.02, HR = 0.39, (95% CI 0.17–0.86)) were significantly related to higher and lower risk of NRM in multivariate analysis.

### 3.3. Incorporation of ELN 2017 Risk Classification and WT1 Level before Allogeneic HSCT

To identify the incorporated prognostic impact of pre-treatment ELN 2017 risk and post-chemotherapy *WT1* levels on clinical outcomes after transplantation, we further divided the patients into the following four subgroups: INT*^WT1^*^low^ (*n* = 82), INT*^WT1^*^high^ (*n* = 26), ADV*^WT1^*^low^ (*n* = 44), and ADV*^WT1^*^high^ (*n* = 22; Figure 3). For patients with *WT1*^low^ and *WT1*^high^ in the INT risk group, the 2 year OS rates were 66.9% vs. 58.70% (*p* = 0.162), RFS rates were 59.5% vs. 21.6% (*p* = 0.009), CIR rates were 12.0% vs. 67.3% (*p* < 0.001), and NRM rates were 32.1% vs. 18.4% (*p* = 0.084), respectively. For those with *WT1*^low^ and *WT1*^high^ in the ADV risk group, the 2 year OS rates were 66.9% vs. 58.7% (*p* = 0.325), RFS rates were 58.9% vs. 43.4% (*p* = 0.189), CIR rates were 19.2% vs. 51.9% (*p* = 0.031), and NRM rates were 13.4% vs. 5.6% (*p* = 0.295), respectively. The importance of high *WT1* transcript levels was more prominent in the INT risk group, showing significant differences in both RFS and CIR between the INT*^WT1^*^low^ and INT*^WT1^*^High^ (Figure 3). Overall, patients with a high *WT1* level at transplant showed worse post-HSCT outcomes in the context of RFS (*p* = 0.017) and CIR (*p* < 0.001) as compared to *WT1*^low^ patients of either the INT or ADV risk group (Appendix A).

### 3.4. Subdivision of ELN Risk Groups: Negative Impact of Mutated FLT3-ITD on Post-HSCT Outcomes Regardless of Allelic Burden or NPM1 Co-Mutation Status

To determine which components constituting the ELN 2017 risk stratification influenced the post-HSCT prognosis the most, we conducted an analysis with these components as variables, where we identified that the presence of the *FLT3*-ITD mutation at diagnosis had an independent negative impact on post-HSCT OS, RFS, and CIR (Appendix A). Next, we focused on the *FLT3*-ITD-mutated group only and performed a detailed analysis according to *FLT3*-ITD allelic burden (*FLT3*-ITD^high^*^,^*
^low^) and *NPM1* mutation status (*NPM1*^wt, mut^), in which we revealed that the subgroup analysis based on these factors did not follow the known ELN 2017 risk outcomes (Figure 4). For the *FLT3*-ITD^high^/*NPM1*^mut^ (*n* = 13), *FLT3*-ITD^low^/*NPM1*^wt^ (*n* = 9), and *FLT3*-ITD^high^/*NPM1^wt^* AML (*n* = 8) groups, the OS and RFS at 2 years after HSCT did not significantly differ (46.2%, 44.4%, and 50% for OS (*p* = 0.795, Figure 4A) and 30.0%, 33.3%, and 36.4% for RFS (*p* = 0.9, Figure 4B), respectively). Moreover, these subpopulations all showed worse OS and RFS when compared to the remaining *FLT3*-ITD-negative patients, belonging to ELN 2017 INT or ADV risk groups. 

### 3.5. Association between FLT3-ITD Mutation and Pre-HSCT WT1 Expression Level

The *WT1* expression levels at pre-HSCT were higher in patients with *FLT3*-ITD mutation than in those with wildtype (mean 0.174 versus 0.067, *p* = 0.028). We evaluated the role of *WT1* expression by applying the *WT1* level within a *FLT3*-ITD mutated subgroup (*n* = 30), where we found that 10 *WT1*^high^ patients showed significantly poor outcomes in OS, RFS, and CIR (*p* = 0.008, *p* = 0.01, and *p* = 0.004, respectively) (Figure 5). Among the 30 patients with *FLT3*-ITD mutation at baseline, 24 patients were tested for pre-HSCT *FLT3*-ITD mutational status using the aforementioned method (PCR for fragment analysis). When receiver operating characteristics (ROC) analysis was used to determine the optimal threshold value of *WT1* level to predict *FLT3*-ITD MRD positivity, the area under the curve (AUC) was 0.7 at the cutoff *WT1* level of 250 copies/10^4^ ABL, with an accuracy of 0.75 in the pre-HSCT samples (Appendix A).

## 4. Discussion

Owing to the recent advances in genomic technologies, AML has been identified as a heterogeneous group of diseases according to cytogenetics and molecular aberrations. This led to substantial changes in the classification and prognostication of AML over the years, and the ELN 2017 classification is the latest updated version of the risk model at present, which can distinguish between significantly different overall survival groups, mainly for those who received chemotherapy-based consolidation.

In this study, we examined the prognostic ability of ELN 2017 risk stratification for post-HSCT outcomes and analyzed a total of 174 AML patients who underwent first allogeneic HSCT as consolidation therapy following myeloablative (79.3%) or reduced-intensity conditioning (20.7%). As allogeneic HSCT was primarily considered for patients in INT or ADV risk groups, we paid particular attention to AML patients in these risk groups, and all the analyzed patients were allocated to either the INT or ADV risk group by ELN 2017, on the basis of their cytogenetic and molecular aberrations at diagnosis. After HSCT, OS at 2 years for the INT and ADV risk groups was 58.6% and 64.4% (*p* = 0.299), respectively, which was not significantly different between the groups. Similarly, the ELN risk at diagnosis was not significantly correlated with post-HSCT outcomes in the context of RFS and CIR. This finding may represent the fact that ELN 2017 risk could not effectively segregate the post-HSCT outcomes of INT risk patients from ADV risk patients. However, it may also suggest that the treatment decision to undergo allogeneic HSCT according to the ELN risk stratification at diagnosis allows for the poor outcomes in patients of the ADV risk group to be overcome.

Our findings conflict with the results of previous reports. A German study [8] and a CIBMTR analysis [42] confirmed that the ELN 2017 classification is applicable to patients receiving HSCT, effectively distinguishing three risk groups with significantly distinct post-HSCT outcomes. When compared to our study, these studies included patients of favorable (FAV) risk in addition to INT or ADV risk groups and focused on the post-transplant outcomes of the whole population rather than focusing only on the INT and ADV risk groups, who are the common candidates for HSCT. In addition, in the German study [8], the distribution of patients at risk was quite disproportional, and the number of INT risk patients was too small to compare with other risk groups; only 30 out of 234 patients (12.8%) were grouped as being INT risk, and the majority were classified as either FAV (*n* = 93, 39.7%) or ADV (*n* = 111, 47.4%). Consequently, the statistical significance was largely due to the difference in outcomes between the FAV and ADV risk groups. Regarding CIBMTR analysis [42], the authors used adapted ELN genetic risk stratification (aELN) for the validation of HSCT outcomes in transplant recipients in AML CR1. In this operational genetic classification, the *FLT3*-ITD allelic ratio and *CEBPA* mono/biallelic status were not used for risk stratification due to a lack of data, and patients with the *FLT3*-ITD mutation were all grouped as ADV risk if the *NPM1* mutation was negative. This potentially led to risk group shifts in patients with *FLT3*-ITD with low allelic mutations; risk groups were changed from FAV (ELN 2017) to INT (aELN) for *FLT3*-ITD^low^/*NPM1*^mut^ cases and from INT (ELN 2017) to ADV (aELN) for *FLT3*-ITD^low^/*NPM1*^wt^ cases, respectively. This is of particular importance given that the decision of whether to undergo HSCT remains a matter of debate in AML with *FLT3*-ITD mutations with a low allelic ratio or concomitant *NPM1* mutation [43,44].

Before the introduction of ELN 2017, the prognostic impact of the ELN 2010 classification for patients receiving allogeneic HSCT was validated by US researchers, where the authors found that ELN ADV and INT1 risk had significantly decreased survival due to an increased risk of post-HSCT relapse [45]. It was notable that the best outcomes in the context of OS and event-free survival (EFS) were in patients with FAV and INT2 risk; INT1 compared to INT2 showed even worse outcomes, with a threefold higher hazard ratio for death. The authors attributed this to the higher percentage of *FLT3*-ITD abnormalities in their INT1 cohort (72%) when compared to the ELN 2010 cohort (20%). The results from this analysis were based on an old version of ELN [4] and, thus, could not provide a detailed prognosis for patients with the *FLT3*-ITD mutation according to the allelic ratio or *NPM1* status. Nevertheless, this may reflect the importance of the *FLT3*-ITD mutation itself for post-HSCT outcomes in AML patients. In line with this result, we confirmed that the *FLT3*-ITD mutation had the greatest impact on post-HSCT outcomes compared to any factor constituting the current ELN 2017 risk stratification. These findings could be the rationale for applying an *FLT3* inhibitor as a maintenance therapy for HSCT patients [46,47,48,49,50,51], and we are awaiting the results of recent clinical trials to determine who would benefit the most from treatment with *FLT3* inhibitors in this setting [52,53].

To identify the significance of the allelic ratio and *NPM1* status in *FLT3*-ITD-mutated AML, the post-HSCT outcomes for patients of different combinations of *FLT3*-ITD^high or low^/*NPM1*^wt or mut^, belonging to INT and ADV risk by ELN 2017, were separately analyzed in this study. As a result, we observed that the outcomes of *FLT3*-ITD-mutated subgroups—*FLT3*-ITD^high^ and *NPM1*^mut^ (*n* = 13), *FLT3*-ITD^low^, and *NPM1*^wt^ (*n* = 9), and *FLT3*-ITD^high^ and *NPM1*^wt^ AML (*n* = 8)—did not significantly differ after transplantation. Moreover, *FLT3*-ITD-mutated subgroups all showed worse post-transplant outcomes when compared to the rest of the patients with INT and ADV risk. Although the number of patients in each group was too small to draw a definitive conclusion, the ELN 2017 classification did not distinguish between different risk groups within *FLT3*-mutated patients, and mutated *FLT3*-ITD regardless of the allelic ratio, or *NPM* status tended to greatly affect post-HSCT outcomes. For treatment decisions for *FLT3*-ITD-mutated patients of different subgroups, our study has a limitation because we did not perform a comparison between patients receiving allogeneic HSCT and those receiving chemotherapy alone. However, a study from the MD Anderson Cancer Center showed that allogeneic HSCT reduced the risk of relapse and improved both RFS and OS, regardless of the *FLT3* allelic ratio and *NPM1* status, as compared to the non-transplant group among the 227 *FLT3*-mutated AML patients [54]. We believe that this finding further emphasizes that *FLT3*-ITD alone could work as an important biologic prognostic factor, even if the adverse impact of this mutation in AML may have some variations depending on other biologic features.

In addition to pretreatment risk stratification, assessment of MRD is also a well-known prognostic factor for AML outcomes. Although the *WT1* transcript level used in this study is less specific and, therefore, has a low priority for MRD assessment in AML, it could effectively discriminate between two groups with different risks of relapse and survival. Of interest, the prognostic impact of the pre-HSCT *WT1* transcript level remained significant even after adjusting for the ELN 2017 classification, while the ELN 2017 risk failed to prove its prognostic significance in both univariate and multivariate analysis. Notably, the pre-HSCT *WT1* level had a more prominent value for outcome prediction in the INT risk group as compared to the ADV risk group. These findings imply that the kinetics and depth of treatment response are just as important as biologic markers at AML diagnosis, and MRD assessment with available assays should be utilized to improve the transplant outcomes by providing a possible guide to determine ideal conditioning intensity [15] or post-HSCT management, such as the rapid tapering of immunosuppression for patients at higher risk of relapse [55].

There are several limitations to the current study. One is the retrospective nature of the study, in addition to the relatively small sample size, including only a few patients in each subgroup for comparison, and the absence of an independent validation cohort, which make it hard to confirm our findings in general. Furthermore, our cohort only included patients who reached CR(i) before transplant. Consequently, adverse risk patients included in this study were highly selected from the beginning and cannot be considered representative of the entire adverse risk group. In addition, although the patients in this study were recruited from November 2017 to November 2020, when the RATIFY [56] and ADMIRAL-trial [57] were already published and midostaurin and gilteritinib became available soon afterward in many countries, it was only much later that these *FLT3* inhibitors became available in Korea. Consequently, no patient included in this study received any of the *FLT3* inhibitors, and we could not evaluate the impact of pre-transplantation treatment with/without *FLT3* inhibitors on post-HSCT outcomes. Lastly, we cannot exclude the possibility that the prognostic value of ELN 2017 in an HSCT setting might be obscured in this study by most patients undergoing MAC transplants. Moreover, the high frequency of haplo-allo-HSCT in this study (up to 43.7%) is likely a unique situation worldwide. Therefore, our findings lack generalizability, and further analysis in a larger cohort is necessary to confirm them.

## 5. Conclusions

In summary, the ELN 2017 risk classification did not effectively predict post-HSCT outcomes in patients in INT and ADV risk groups. The pre-HSCT *WT1* level, rather than ELN 2017, was superior in predicting post-HSCT relapse, and it had a more prominent prognostic value in the ELN INT risk group than the ADV risk group. The *FLT3*-ITD mutation had the greatest impact on post-HSCT outcomes among the components constituting ELN 2017 risk stratification, while *FLT3*-ITD-mutated subgroups showed the worst post-HSCT outcomes regardless of allelic ratio or *NPM1* status when compared to the other patients at INT and ADV risk.

## Figures and Tables

**Figure 1 cancers-14-03199-f001:**
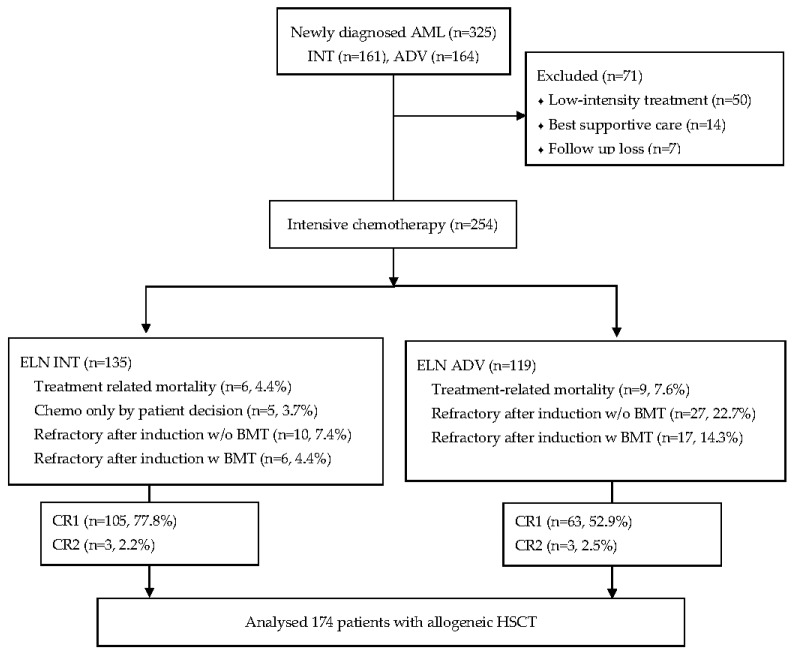
CONSORT flow diagram for patient selection.

**Figure 2 cancers-14-03199-f002:**
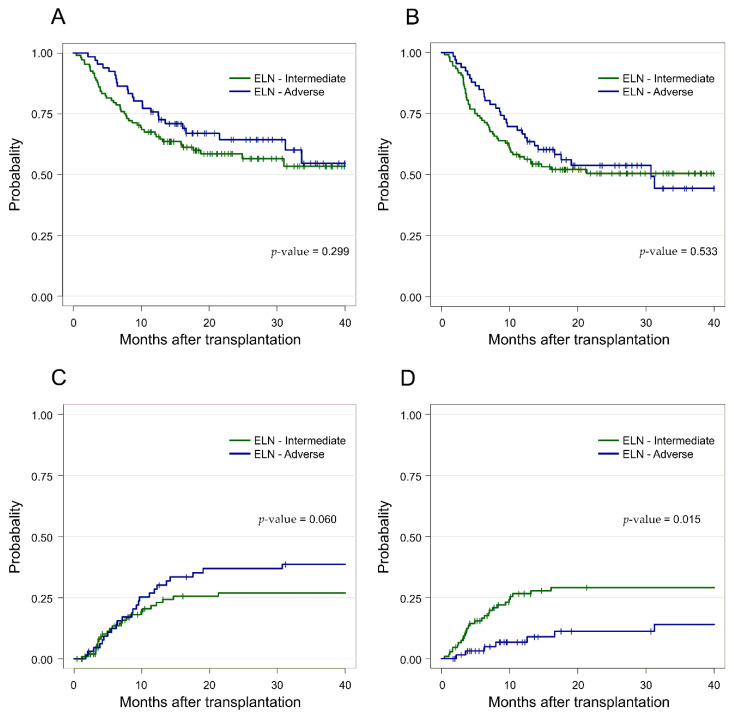
Prognostic significance of the European Leukemia Net (ELN) 2017 risk classification in acute myeloid leukemia patients who had undergone allogeneic hematopoietic stem-cell transplantations. The ELN 2017 classification predicted the (**A**) overall survival (OS), (**B**) relapse-free survival (RFS), (**C**) cumulative incidence of relapse (CIR), and (**D**) non-relapse mortality (NRM) of patients belonging to the two risk groups.

**Figure 3 cancers-14-03199-f003:**
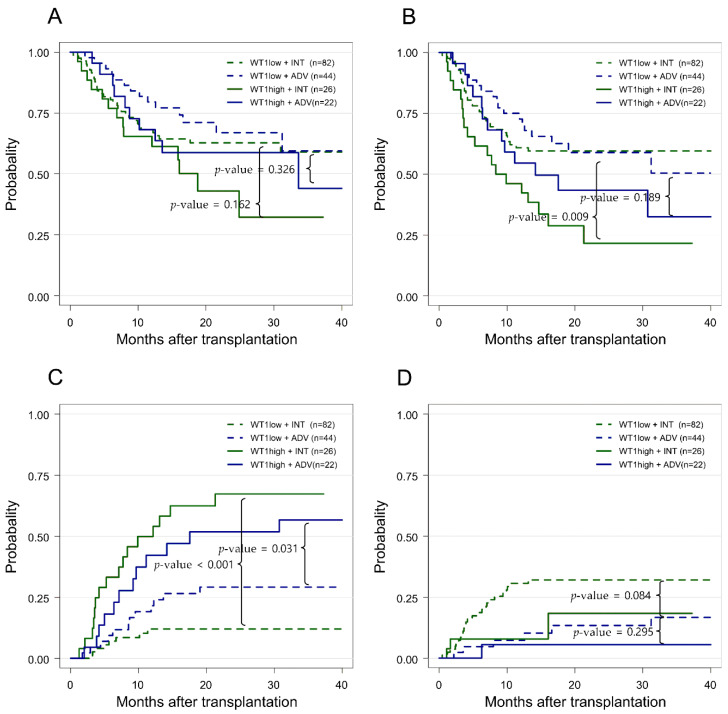
Prognostic significance of Wilms tumor gene 1 (*WT1)* transcript levels for post-transplantation outcomes in terms of the European Leukemia Net 2017 classification components: (**A**) overall survival (OS); (**B**) relapse-free survival (RFS); (**C**) cumulative incidence of relapse (CIR); (**D**) non-relapse mortality (NRM). WT1low + INT, intermediate risk patients with low *WT1* levels; WT1low + ADV, adverse risk patients with low *WT1* levels; WT1high + INT, intermediate risk patients with high *WT1* levels; WT1high + ADV, adverse risk patients with high *WT1* levels.

**Figure 4 cancers-14-03199-f004:**
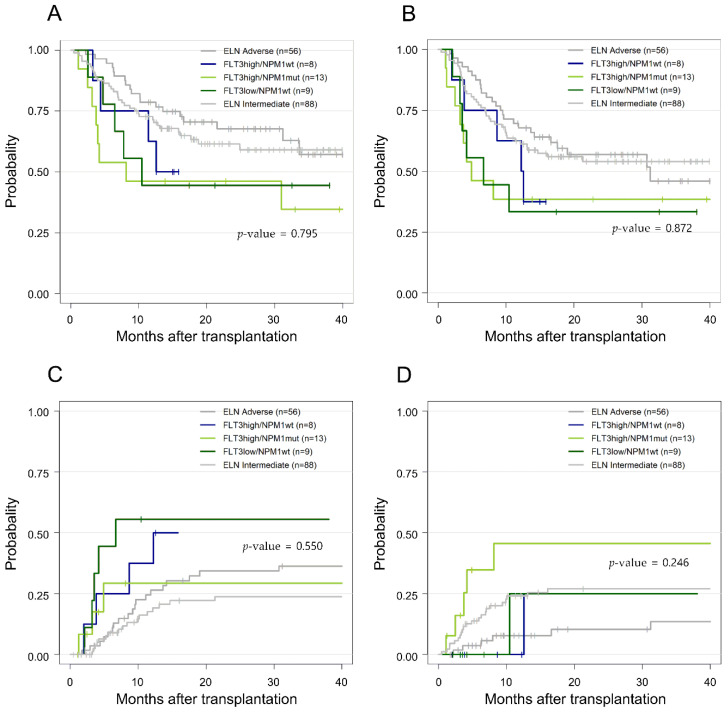
Comparison of (**A**) overall survival (OS), (**B**) relapse-free survival (RFS), (**C**) cumulative incidence of relapse (CIR), and (**D**) non-relapse mortality (NRM) in acute myeloid leukemia patients with FMS-like tyrosine kinase 3 internal tandem duplication mutation, sub-grouped as per the European Leukemia Net (ELN) 2017 classification. ELN Adverse, adverse risk patients; ELN Intermediate, intermediate risk patients; FLT3high/NPM1mut, patients with high *FLT3* levels and mutated *NPM1*; FLT3high/NPM1wt, patients with high *FLT3* levels and wildtype *NPM1*; FLT3low/NPM1wt, patients with low *FLT3* levels and wildtype *NPM1*.

**Figure 5 cancers-14-03199-f005:**
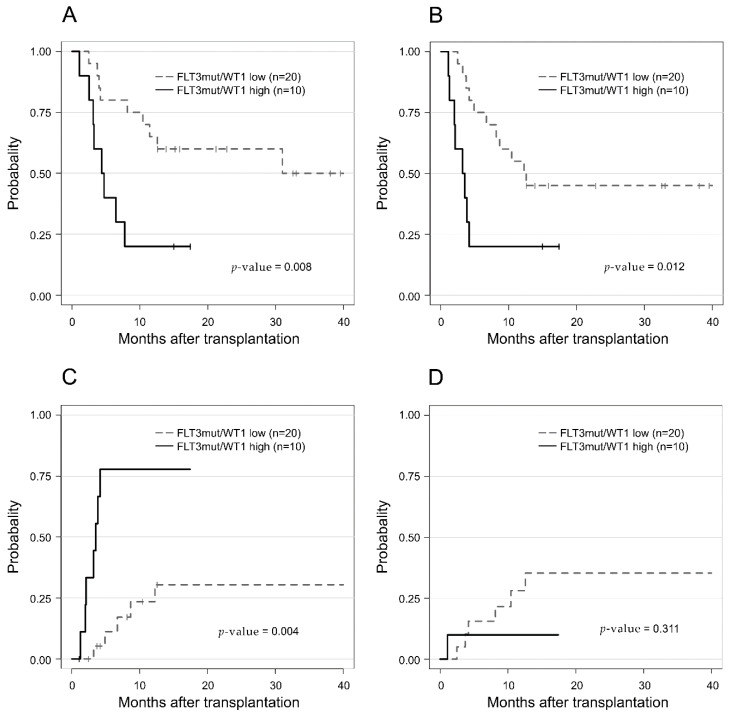
Comparison of (**A**) overall survival (OS), (**B**) relapse-free survival (RFS), (**C**) cumulative incidence of relapse (CIR), and (**D**) non-relapse mortality (NRM) in acute myeloid leukemia patients with the *FLT3*-ITD mutation, divided by pre-HSCT *WT1*^high^ and *WT1*^low^.

**Table 1 cancers-14-03199-t001:** Baseline demographic and clinical characteristics of the study cohort (n = 174).

Characteristics, Number of Patients (%)	All Patients	Intermediate-Risk Patients (*n* = 108)	Adverse-Risk Patients (*n* = 66)	*p*-Value
Age at diagnosis, years and range	54 (18–72)	54 (18–74)	53 (20–72)	0.759
<55 years	91 (52.3%)	55 (50.9%)	36 (54.5%)	
≥55 years	83 (47.7%)	53 (49.1%)	30 (45.5%)	
Gender				0.480
Male	85 (48.9%)	50 (46.3%)	35 (53.0%)	
Female	89 (51.1%)	58 (53.7%)	31 (47.0%)	
HCT-CI				0.621
0–2	108 (62.1%)	65 (60.2%)	43 (65.2%)	
≥3	66 (37.9%)	43 (39.8%)	23 (34.8%)	
Disease type				0.642
De novo AML †	160 (92.0%)	98 (90.7%)	62 (93.9%)	
Secondary AML	14 (8.0%)	10 (9.3%)	4 (6.1%)	
Conditioning intensity				0.047
Myeloablative	138 (79.3%)	80 (74.1%)	58 (87.9%)	
Reduced Intensity	36 (20.7%)	28 (25.9%)	8 (12.1%)	
Donor type				0.962
MSD	45 (25.9%)	28 (25.9%)	17 (25.8%)	
MUD	46 (26.4%)	28 (25.9%)	18 (27.3%)	
HID	76 (43.7%)	47 (43.5%)	29 (43.9%)	
Others (UCB or MMUD)	7 (4.0%)	5 (4.6%)	2 (3.0%)	
Complete remission				0.848
CR1	168 (96.6%)	105 (97.2%)	63 (95.5%)	
CR2	6 (3.4%)	3 (2.8%)	3 (4.5%)	
*WT1* level ^§^ at pre-HSCT				0.250
Low	126 (72.4%)	82 (75.9%)	44 (66.7%)	
High	48 (27.6%)	26 (24.1%)	22 (33.3%)	

AML, acute myeloid leukemia; MSD, matched sibling donor; MUD, matched unrelated donor; MMUD, mismatched unrelated donor; HID, haploidentical donor; UCB, umbilical cord blood; ELN, European Leukemia Net; HCT-CI, hematopoietic stem-cell transplantation-specific comorbidity index; HSCT, hematopoietic stem-cell transplantation; † de novo AML encompasses no clinical history of prior myelodysplastic syndrome (MDS), myeloproliferative disorder, or exposure to potentially leukemogenic therapies or agents; ^§^ *WT1* cutoff level of 250 copies per 10^4^
*ABL* to determine *WT1*^high^ and *WT1*^low^.

**Table 2 cancers-14-03199-t002:** Univariate and multivariate analyses of the survival outcomes.

		Univariate		Multivariate	
End Point and Variable I (Total *n* = 174)	*N*	HR (95% CI)	*p*-Values	HR (95% CI)	*p*-Values
**OS**					
Age ≥55 years	60	1.65 (1.02–2.65)	0.039	1.60 (0.97–2.63)	0.065
ELN adverse risk group	66	0.77 (0.47–1.27)	0.299	0.70 (0.43–1.16)	0.171
Secondary AML	14	0.66(0.24, 1.81)	0.388		
Conditioning intensity, RIC	36	1.11 (0.63–1.96)	0.727		
Donor type, HID	76	1.69 (1.05–2.75)	0.029	1.62(0.99–2.64)	0.055
CR2	6	2.22 (0.81–6.11)	0.121		
High *WT1* levels	48	1.51 (0.92–2.48)	0.110	1.81(1.09–3.02)	0.022
**RFS**					
Age ≥55 years	60	1.29 (0.84–1.98)	0.249		
ELN adverse risk group	66	0.87 (0.56–1.36)	0.533	0.79 (0.50–1.23)	0.296
Secondary AML	14	1.01 (0.46–2.19)	0.984		
Conditioning intensity, RIC	36	1.04 (0.61–1.78)	0.881		
Donor type, HID	76	1.41 (0.92–2.17)	0.118		
CR2	6	2.40 (0.97–5.93)	0.058	1.84 (0.73–4.64)	0.198
High *WT1* levels	48	2.06 (0.83–5.10)	0.003	1.92 (1.21–3.03)	0.005
**CIR**					
Age ≥55 years	60	0.46 (0.25–0.84)	0.012		
ELN adverse risk group	66	1.71 (0.98–3.01)	0.060	1.32 (0.73–2.36)	0.360
Secondary AML	14	1.38 (0.56–3.42)	0.488		
Conditioning intensity, RIC	36	0.64 (0.28–1.46)	0.291		
Donor type, HID	76	0.8 (0.45–1.41)	0.440		
CR2	6	5.27 (2.20–12.6)	0.005	2.67 (1.20–5.94)	0.016
High *WT1* level	48	4.85 (2.74–8.59)	<0.001	4.22 (2.28–7.81)	<0.001
**NRM**					
Age ≥55 years	60	1.62 (2.19–11.6)	<0.001	3.90 (1.66–9.15)	0.002
ELN adverse risk group	66	0.36 (0.16–0.82)	0.015	0.39 (0.17–0.86)	0.020
Secondary AML	14	0.66 (0.16–2.70)	0.564		
Conditioning intensity, RIC	36	1.61 (0.78–3.34)	0.196		
Donor type, HID	76	2.45 (1.25–4.83)	0.009	1.76 (0.92–3.39)	0.090
CR2	6	-	>0.999		
High *WT1* levels	48	0.31 (0.11–0.89)	0.030	0.46 (0.15–1.40)	0.174

ELN, European Leukemia Net; AML, acute myeloid leukemia; CI, confidence interval; RFS, relapse-free survival; HR, hazard ratio; OS, overall survival; TRM, treatment-related mortality; CIR, cumulative incidence of relapse; NRM, non-relapse mortality; HID, haploidentical donor; RIC, reduced-intensity conditioning; N, number of patients.

## Data Availability

The data presented in this study are available on request from the corresponding author. The data are not publicly available owing to privacy and ethical reasons.

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
