# Peer review of "Depth of Response to Intensive Chemotherapy Has Significant Prognostic Value among Acute Myeloid Leukemia (AML) Patients Undergoing Allogeneic Hematopoietic Stem-Cell Transplantation with Intermediate or Adverse Risk at Diagnosis Compared to At-Risk Group According to European Leukemia Net 2017 Risk Stratification"

_cancers, 2022, doi:10.3390/cancers14133199_

Round 1
Reviewer 1 Report
Although I still find some English typos and grammatical mistakes which should be corrected. So I recommend English Editing.
The authors made changes in blue color without a file with their reply point by point.
So this file should be included to review point by point.

Author Response
We received an English editing service. We attached the point-to-point review included in the cover letter. Thank you for showing interest in our study.

Reviewer 2 Report
I agree with the modified version of the manuscript.
Author Response
We received an English editing service. Thank you for your positive feedback on our study.

Reviewer 3 Report
I acknoledge that the authors put effort in the revisions and provide detailed answers to my remarks. However, a few points remain:
Title: I really don't get, why FLT3-ITD mutational status is now included in the title. First, patients in the study were not treated according to today's standard of care (FLT3i), thus respective concusions might be outdated. And second: I understand that this paper is meant to report about the status of WT1 expression levels. Why then talk about FLT3 in the title?
Author Response
Thank you for your deep-intergreted feedback on our study. We agree with your comments, and decide to change our title to "Depth of response to intensive chemotherapy rather than risk group at diagnosis has significant prognostic value among acute myeloid leukemia (AML) patients with intermediate or adverse risk according to European Leukemia Net2017 risk stratification undergoing allogeneic hematopoietic stem cell transplantation". Moreover, we received an english editing service.

This manuscript is a resubmission of an earlier submission. The following is a list of the peer review reports and author responses from that submission.
Round 1
Reviewer 1 Report
Dear Editor
Appreciated thanks for your invitation to act as a reviewer in your respected Journal Cancers
I would like to accept this article after only minor English editing, the authors represented the prognostic value of the ELN 2017 criteria on post-hematopoietic stem cell transplantation (HSCT) 23
outcomes and compared it with pre-HSCT measurable residual disease (MRD) status determined 24 by Wilms tumor gene 1 (WT1) expression, the authors classified the patients as intermediate (INT) risk and 25
adverse (ADV). We found that the ELN 2017 risk classification did not effectively predict post-HSCT 26 outcomes in patients with INT and ADV-risk groups. Pre-HSCT WT1 level predicted post-HSCT 27 relapses better than ELN 2017, and it had a more prominent prognostic value in the ELN INT-risk 28
group rather than in the ADV-risk group. During revision. I find them describe their discussion in an appropriate way. upon that, I consider this manuscript as accepted after minor English revision and rewrite the methodology to explore the depth of the study
Reviewer 2 Report
The study is rigorously conducted and provides important information for clinicians. I recommend publication in the actual form.
Reviewer 3 Report
In this study by Kim et al., the prognostic impact of WT1 transcription levels prior to allogeneic transplantation is evaluated and compared to ELN2017 risk classification in 174 intermediate and adverse risk AML patients. The manuscript is well written and organized, however, there are major weaknesses of the study.
Major:
1) The title is misleading. It is not the purpose of the ELN2017 risk classification to predict post HSCT outcomes. This, however, is indirectly implied in the title. Need to change it.
2) WT1 as MRD-marker: There is no explanation in the introduction about the scientific background of WT1 expression as MRD marker
3) Though there might be differences between countries/continents: I am not aware of institutions in my and other countries that use WT1 expression in routine diagnostics. Please present rough numbers to which extend this is common practice in which areas of the world.
4) The text states that all patients received induction therapy with cytarabin and anthracyclin. However, a substantial part of patients was transplanted in CR2. What was given as second-line therapy?
5) The aspect that I dislike most about the paper is the fact that there are extensive elaborations on the adverse impact of FLT3, but the impact of FLT3-inhibition is nowhere evaluated. Moreover, the existence of FLT3inhibitors is completely ignored, as they aren't even discussed. According to the methods, patients were recruited form Nov 2017 onwards, when the RATIFY-trial was already published and Midostaurin became available soon afterwards. Therefore, the pre-transplantation treatment with/without FLT3i (Midostaurin or others) urgently needs to be evaluated and taken into statistical consideration.
6) on the same note: did WT1 expression correlate with FLT3 mutational status?
7) Almost half of the donors were haploidentical donors. This is extremely unusal and might have significant impact on the study design. Though choice of donors might vary accross the world, this needs to be evaluated and discussed.
8) I suppose that in addition to WT1 other MRD markers were used at least in some patients (e.g. NPM1)? How do these correlate with WT1 expression status?
Minor:
1) Line 60-62. This sentence seems to be out of context? Copy/paste error?
2) Line 74: Provide citations for this statement. As this is one of the key statements in the introduction, as it justifies the study design, it really makes me doubtful about the impact of WT1 if no citations are provided.
3) Publication of ELN2010 guidelines is now 12 years past. Maybe, we will soon have even a new classification. Why waste so much words on something that's outdated? Could instead use the space to elaborate on FLT3-inhibitors.
4) A CONSORT diagram should be included.